# Functional and Anatomical Micro-Structural Recovery of Idiopathic Macular Holes Following the Inverted Internal Limiting Membrane Flap Technique: A Long-Term Study

**DOI:** 10.3390/diagnostics15232961

**Published:** 2025-11-22

**Authors:** Kai-Ling Peng, Ya-Hsin Kung, Tsung-Tien Wu

**Affiliations:** 1Department of Ophthalmology, Kaohsiung Veterans General Hospital, 386, Ta-Chung 1st Road, Kaohsiung 813, Taiwan; klpeng@vghks.gov.tw (K.-L.P.); yhkung@vghks.gov.tw (Y.-H.K.); 2Shu-Zen Junior College of Medicine and Management, Kaohsiung 821, Taiwan; 3School of Medicine, National YangMing Chiao Tung University, Taipei 112304, Taiwan

**Keywords:** fluid–gas exchange, idiopathic macular hole, inverted internal limiting membrane flap technique, vitrectomy

## Abstract

**Background**: Idiopathic macular holes (MHs) are typically treated with pars plana vitrectomy and internal limiting membrane (ILM) peeling. The inverted ILM flap (ILMF) technique has emerged for MHs, but long-term outcome data remain inadequately established. This study evaluates the long-term functional and anatomical outcomes of the ILMF in idiopathic MHs. **Methods**: We evaluated 71 consecutive eyes of patients with idiopathic MHs who underwent vitrectomy with the inverted ILMF. Follow-up duration was more than 12 months. Visual acuity was measured, and macular anatomy was monitored with optic coherence tomography (OCT). Long-term visual and anatomical outcomes were defined a priori and analyzed accordingly. **Results**: Final vision values showed significant improvement compared to preoperative ones, from 1.02 [Snellen Equivalent (SE), 19/200] ± 0.40 logarithm of the minimum angle of resolution (logMAR) to 0.47 (SE, 68/200) ± 0.39 logMAR (*p* < 0.001). The primary MH closure rates were 94.37% (67/71), while the secondary closure rate reached 97.18% (69/71). Factors associated with better final vision included smaller hole size, favorable hole stage, better preoperative vision, intact postoperative foveal microstructure and contour. The recovery of the external limiting membrane (ELM), inner and outer segment junction (IS/OS), and good foveal contour had improved to 73.4%, 40.3%, and 49.3% at one year and 80%, 71.4%, and 53.3% at three years postoperatively, respectively. **Conclusions**: In idiopathic MHs, the ILMF approach provides meaningful, long-term visual and microstructural recovery, especially with a favorable functional outcome and intact postoperative microstructure sustaining up to three years.

## 1. Introduction

A full-thickness macular hole (FTMH) is defined as a complete defect in the foveal retina layer, and it was initially described in 1869 by Knapp et al. [1] A staging system for FTMHs was proposed in 1995 by Gass et al. [2,3,4]. Today, spectral-domain optical coherence tomography (SD-OCT) is the diagnostic gold standard, allowing for monitoring of the foveal microstructure and vitreomacular interface before and after treatment/surgery [5,6,7,8]. The International Vitreomacular Traction Study Group (IVTS) proposes that macular holes arise from an initial vitreomacular adhesion with anomalous posterior vitreous detachment, generating tangential traction that leads to foveal dehiscence and hole formation [9].

In 1991, vitrectomy was first introduced as an effective macular hole (MH) treatment for traction removal and stimulation of the surrounding tissue repair [10]. Kurt et al. compared internal limiting membrane (ILM) peeling with ILM non-peeling and concluded that peeling is superior for patients with idiopathic stage 2, 3, and 4 MHs, [11] a surgical approach originally proposed by Echardt et al. [12]. In 2009, Michalewska et al. [13] first described the inverted ILM flap (ILMF) technique for difficult MH cases such as those with a large hole, [14] persistent holes, or high-myopia MHs [15]. Inverted ILMFs placed over MHs serve as bridges between the MH edges, facilitating Müller-cell migration and gliosis [16]. This process occurs in a dry environment with the ILM acting as a scaffold separating the hole from the vitreous fluid [17]. Thin ILMFs covering the MH after surgery aid in retinal reconnection and foveal restoration, starting with the external limiting membrane (ELM) and progressing to the inner and outer segment junction (IS/OS) [18].

The exact size at which MHs are capable of closure after ILM peeling or inverted ILMF cannot be precisely defined. If an MH fails to close after ILM peeling, techniques such as an ILM or posterior capsular free flaps, retinal autografts, or amniotic membrane grafts may be applied. In cases of larger, more challenging MHs [14,15], the inverted ILMF has gradually replaced ILM peeling as the standard treatment supported by meta-analytic evidence suggesting a higher MH closure rate [19]. Herein, we evaluated long-term functional visual outcomes, anatomical MH closure, and restoration of the foveal microstructure using inverted ILMF techniques across idiopathic FTMHs of different sizes.

## 2. Materials and Methods

### 2.1. Inclusion Criteria and Exclusion Criteria

This retrospective study adhered to the tenets of the Declaration of Helsinki and was approved by the Institutional Review Board of Kaohsiung Veterans General Hospital (number 20-CT6-08, 14 May 2025). We reviewed medical records of consecutive patients who underwent pars plana vitrectomy using the superiorly inverted ILMF technique for idiopathic MH between June 2015 and November 2018. The inclusion criteria included idiopathic MHs with axial lengths (AL) < 26.0 mm and ≥1-year postoperative follow-up. The exclusion criteria included combined with cataract surgery; follow-up >1 year; a history of ocular trauma or vitreoretinal surgery; and concomitant retinopathy or vision-affecting conditions [e.g., diabetic retinopathy, choroidal neovascularization, uveitis, or retinal detachment (RD)].

### 2.2. Primary Outcomes, Secondary Outcomes and Data Collections

Primary outcome: closure of the macular hole at the final follow-up visit. Secondary outcomes: improvement in best-corrected visual acuity (BCVA) and restoration of macular microstructure as assessed by SD-OCT (ELM, IS/OS integrity, and foveal contour). Best-corrected visual acuity (BCVA) was converted to the logarithm of the minimum angle of resolution (logMAR) for analysis. SD-OCT (RTVue scanner; Optovue, Inc. Fremont, CA, USA) was used to evaluate the MH microstructure. Multiple scans (a scan length of 6 or 9 mm) were centered on the fovea. Intraretinal cysts were classified as outer cysts (round hyporeflective spaces extending to the outer plexiform layers) or inner cysts (long ovoid hyporeflective spaces in the nuclear layers) (Figure 1a). MH sizes were measured parallel to the retinal pigment epithelium (RPE) at the closest point of retinal apposition and staged according to Gass et al.: 1, impending foveal depression loss; 2, full-thickness retinal defect with a pseudo-operculum connection; 3, full-thickness retinal defect with persistent hyaloid attachment; or 4, full-thickness retinal defect with complete posterior vitreous detachment [2,3,4]. In stages 1 and 2, the closest point between the two ends of the retinal tissue below the pseudo-operculum was measured.

An ILMF covering the MH indicates flap closure, a type of MH closure (Figure 1b) [20,21]. Gap closure and opening were not included in the MH closure criteria. Homogenous hyper-reflective glial cell proliferation with ILMFs in the foveal area on OCT was termed glial proliferation (Figure 1c,d), whereas ILMFs with thinning hyper-reflective glial cell proliferations were termed foveal thinning (Figure 1e,f). Subretinal fluid (SRF) (Figure 1g) may appear temporary postoperatively. Complete MH closure was defined as a clear foveal layer without glial cell proliferation or residual ILMF, exhibiting a good foveal contour (Figure 1h).

### 2.3. Surgical Procedure

All patients underwent a 23- or 25-gauge pars plana vitrectomy with subsequent ERM, ILM peeling and tamponade using 16% perfluoropropane gas. As Brilliant Blue G was unavailable in Taiwan during June 2015–November 2018, we used 0.05% Indocyanine Green (ICG) for ILM staining, informing patients of possible toxicities. The inverted ILMF technique involves peeling the ILM while preserving a residual superior flap attached to the MH edges, effectively spanning the hole rather than tucking into it. A suitably sized superior inverted flap was placed over the MH and secured in place with intraocular forceps. During fluid–gas exchange, a backflush needle was used to aspirate fluid near the inferior arcade to facilitate flow and help maintain the ILMF covering of the MH.

### 2.4. Data Analysis

Two groups were created according to MH size: <200 µm and ≥200 µm. Independent *t*-tests were used to compare categorical variables. Pre- and postoperative visions were compared using paired *t*-tests. Chi-square tests were used to compare categorical variables. Data were analyzed using IBM SPSS statistical software (version 20.0; Armonk, NY, USA). *p* < 0.05 denotes statistical significance.

## 3. Results

### 3.1. General Data Analysis and Primary Outcomes

During the study period, 311 eyes with various diseases were operated on by a single retinal surgeon during the study period. Among these, 237 eyes underwent the inverted ILMF technique, from which we excluded 43 eyes with high myopia (AL > 26.0 mm), 11 with MHs secondary to causes such as diabetic retinopathy, 3 with branch retinal vein occlusion, 3 with trauma, 8 with rhegmatogenous RD, 1 with retinitis pigmentosa, 2 with choroidal neovascularization, 14 with a history of previous vitreoretinal surgery, and 1 with corneal scarring. Finally, 151 eyes with idiopathic MHs were assessed, of which 71 were followed up for at least one year postoperatively.

The mean AL was 23.65 ± 0.98 mm (range, 21.53–24.94; median, 23.44) with a mean refractive error of −0.16 ± 0.28 Diopters (range, −5.75 ± 4.25; median, 0.00). The mean age was 62.38 ± 6.94 years (range, 47–77; median, 62) and 80.28% (57/71) were women. The primary closure rate was 94.37% and the mean follow-up was 20.59 ± 9.87 months (range, 12–59 months; median, 17). Table 1 summarizes the baseline characteristics of the study population. The mean preoperative vision was 1.02 [Snellen Equivalent (SE), 19/200] ± 0.40 logMAR, and the mean final vision was 0.47 (SE, 68/200) ± 0.39 logMAR, with significant improvement observed between preoperative and final visions (*p* < 0.001).

One month after surgery, eight eyes (11.27%) showed ILMF closure without connecting the inferior retinal tissue, and six (8.45%) showed MH non-closure. Regarding ILMF closure alone, six eyes (8.45%) showed complete MH closure two months postoperatively. The other two required another fluid–gas exchange (FGE) and achieved complete MH closure. Regarding poor MH closure one month postoperatively, two eyes slowly closed three months and one year postoperatively with the same final vision (2/20); one underwent another FGE with better final vision (12/20) and two retained MHs with poor final vision (2/20, 6/200). The last eye underwent amniotic membrane transplantation to achieve MH closure and a final vision of 1/20. The primary and secondary MH closure rates were 94.37% (67/71) and 97.18% (69/71), respectively. Both cases with a persistent MH were in the ≥200 µm MH group, with a mean MH size of 276.28 ± 9.97 µm. No complications were observed.

Preoperative factors affecting final vision included the hole size [*p* < 0.001, correlation coefficient (CC) = 0.417] and preoperative vision (*p* < 0.001, CC = 0.356). Postoperative factors influencing final vision included hole non-closure at one month postoperatively, final continuous ELM and IS/OS, final good foveal contour (*p* < 0.001), and the follow-up time (*p* < 0.001, CC = −0.446).

### 3.2. Comparison of MHs of Different Sizes (<200 µm vs. ≥200 µm)

The two groups of <200 µm and ≥200 µm differed significantly in ILMF trapped in MH or touched RPE (*p* = 0.022), ILMF closure one month postoperatively (*p* = 0.026), final continuous ELM (*p* = 0.009) and IS/OS (*p* = 0.009), good final foveal contour, and final visions (*p* < 0.001). Vision improvement was significant in both groups (<200 µm, *p* < 0.001; ≥200 µm, *p* < 0.001). Table 2 summarizes the baseline demographic data and preoperative/postoperative characteristics of the two groups.

### 3.3. Secondary Outcomes: Functional Recovery

Figure 2 shows the changes in the mean BCVA logMAR of the different groups over the follow-up period. The mean BCVA in the <200-µm MH-size group significantly improved from 0.95 (SE, 28/250) ± 0.35 logMAR at baseline to 0.69 (SE, 41/200) ± 0.32 logMAR at three months (*p* < 0.001, paired *t*-test), mildly declined to 0.81 (SE, 48/200) ± 0.47 logMAR at six months, and significantly improved to 0.55 (SE, 56/200) ± 0.45 logMAR at one year (*p* < 0.001), and then 0.32 (SE, 96/200) ± 0.24 logMAR at two years (*p* = 0.01). Finally, it slightly improved to 0.2 (SE, 126/200) ± 0.04 logMAR at three years. However, the mean BCVA in the ≥200 µm MH group slightly improved from 1.09 (SE, 16/200) ± 0.44 logMAR at baseline to 0.94 (SE, 29/250) ± 0.39 logMAR at three months, mildly declined to 1.0 (SE, 20/200) ± 0.48 logMAR at six months, slightly improved to 0.85 (SE, 35/250) ± 0.35 logMAR at one year, statistically improved to 0.51 (SE, 62/200) ± 0.34 logMAR at two years (*p* = 0.014) and slightly improved to 0.32 (SE, 96/200) ± 0.26 logMAR at three years. In total, the mean BCVA statistically improved from 1.02 (SE, 19/200) ± 0.40 logMAR at baseline to 0.81 (SE, 31/200) ± 0.37 logMAR at three months (*p* < 0.001), and then mildly declined to 0.90 (SE, 31/250) ± 0.48 logMAR, but it significantly improved to 0.70 (SE, 40/200) ± 0.43 logMAR at one year and 0.41 (SE, 78/200) ± 0.30 logMAR at two years (*p* < 0.001, respectively) and slightly improved to 0.27 (SE, 107/200) ± 0.20 logMAR at three years. There was a significant difference in mean postoperative BCVA differed significantly between the two groups at three months postoperatively (*p* = 0.005), in vision one year postoperatively (*p* = 0.004), and in final vision (*p* < 0.001).

### 3.4. Secondary Outcomes: Micro-Structural Recovery

Figure 3 shows the microstructural recovery of the study population at a three-year follow-up. ILMF closure, MH non-closure, and SRF status occurred in 11.3%, 4.3%, and 1.4%; 8.5%, 5.8%, and 5.8%; 2.9%, 12.7% and 7.0%; and 4.2% and 2.8%, at one, three, and six months and one year postoperatively, respectively (Figure 3A). Within three years, a complete foveal contour was seen in for 36.6–53.3%, glial proliferation in 11.3–13.3%, and foveal thinning in 19.7–33.3% (Figure 3B). Discontinuous ELM was seen in 64.2% (≥200 µm MH group, 34.3%; <200 µm MH-group, 29.9%), 41.5% (≥200 µm MH group, 27.7%; <200 µm MH group, 13.8%), 34.4% (≥200 µm MH group, 24.6%; <200-µm MH group, 9.8%), 26.6% (≥200 µm MH-group, 18.8%; <200 µm MH group, 7.8%), 16.1%, and 20.0% of the ≥200 µm MH group at one, three, and six months, and one, two, and three years postoperatively (Figure 3C). The respective values for discontinuous IS/OS were 94.1% (≥200 µm MH group, 42.6%; <200 µm MH group, 51.5%), 83.1% (≥200 µm MH group, 40.4%; <200 µm MH group, 43.1%), 71% (≥200 µm MH group, 38.7%; <200 µm MH group, 32.3%), 59.7% (≥200 µm MH group, 35.8%; <200 µm MH group, 23.9%), 54.7% (≥200 µm MH group, 32.2%; <200 µm MH group, 22.5%), and 28.6% (≥200 µm MH group, 28.6%; <200 µm MH group, 0%) (Figure 3C). Overall, ILMF closure and MH non-closure were resolved within six months and one year, respectively (Figure 3A). However, glial proliferation remained in 11.3–13.3%, with an increase in foveal thinning in 9.1–15.8% and an increase in good foveal contour in 1.4–16.7% at two years postoperatively, with these levels maintained at three years (Figure 3B). Interrupted ELM was resolved in the <200µm MH group within one year, while it was still an issue in 16.1–20.0% of the ≥200 µm group (Figure 3C). Discontinuous IS/OS was resolved in 51.5% of the <200 µm MH group at two years postoperatively but only in 14% of the ≥200 µm MH group within three years (Figure 3D).

## 4. Discussions

Two different inverted ILMF techniques have been proposed: the classic inverted ILMF technique in which the ILM around the MH is peeled 360° to its edges and trimmed to cover the MH [18,20,22] and the temporal ILMF technique where the temporal ILMF is peeled to the MH margin and inverted to cover the MH without peeling the residual ILM around the upper/lower/nasal retina [18,20,23,24,25]. Several studies [26,27,28,29,30] have used an inverted ILMF but peeled all the other ILMs to the arcades to leave an ILMF of an appropriate size to cover the MH. The primary MH closure rates achieved using the classic techniques were 100% (125 eyes; mean MH size, 494.78 µm) [18], 93% (40/43; mean MH size, 533 µm; range 400–763 µm) [20], and 91.93% (320 eyes in total) [22]. The inverted ILMF technique without peeling achieved rates of 93% (41/44; mean MH size, 544 µm; range 400–841 µm) [20], 88.9% (16/18; mean MH size, 577.4 µm) [23], 98% (54/55; MH size, <400 µm) [24], and 90.5% (38/42; MH size, >650 µm) [25]. Finally, the inverted ILMF technique with peeling achieved rates of 98.53% (67/68; mean MH size, 560 µm) [26], 94.8% (53 eyes; mean MH size, 592 µm) [27], 100% (24 eyes; mean MH size, 282 ± 104 µm) [28], and 100% (100 eyes; mean MH size, 198.9 ± 30.4 µm and 328.4 ± 46.3 µm) [29]. Using these three different ILMF techniques, the primary closure rate reached >90% regardless of MH sizes. Studies with larger MHs have also used perflucarbone, viscoelastic material, or both to firmly settle the ILMF to achieve higher MH closure rates [24]. The MH closure rates with ILM insertion are 97.6% (41/42; mean MH size, 764.12 ± 78.63 µm) and 100% (15/15; mean MH sizes, 662.1 ± 120.9 µm), showing a high MH closure rate in larger MHs [31]. Here, we used superior inverted ILMFs with all ILM peeling to the arcade under gas tamponade without adjuvant assistance. Our primary closure rate was 94.37% (67/71), in line with previous studies while the secondary MH closure rate reached 97.18% (69/71, mean MH size: 181.70 ± 94.56 µm). The two cases with a persistent MH were both in the ≥200-µm MH-group with a mean MH size of 276.28 ± 9.97 µm. Larger MHs were significantly related to ILMF trapped in the MH or touching the RPE one month postoperatively (*p* = 0.022), ILMF closure (*p* = 0.026), poor final vision (*p* < 0.001, CC = 0.417), and poor final foveal microstructures including ELM disruption (*p* = 0.009), IS/OS discontinuity (*p* = 0.008), and abnormal foveal contour (*p* < 0.001).

Previous studies have used ILMF closure [20] and closed MHs with SRF during the early transitional period. The flap-closure rate was 3% (mean MH size, 533 µm; range 400–763 µm) [20] compared to 17.86% [18] using the classic inverted ILMF technique at one year postoperatively, 6% (mean MH size, 544 µm; range 400–841 µm) at three months postoperatively [20], and 15.05% (mean MH size, 494.78 µm) at one year postoperatively using the temporal ILMF technique without residual ILM peeling [18]. They concluded that flap closure occurred in cases with larger MHs [18], which agrees with our results. We further found that FGE may promote inferior foveal tissue connection and the total resolution of flap closure within ten months postoperatively without any deterioration in final visions [32]. The other specific finding of foveal detachment, MH closure with SRF, was 30% after temporal ILMFs without residual ILM peeling one month postoperatively, and it disappeared within six months postoperatively [24]. This accounted for 12.68% (9/71) one month postoperatively in our study which did not affect final visions or lead to significant differences between MH sizes. This may be completely resolved within two years postoperatively.

In the study by Michalewska et al. (MH size, >400 µm), classic and temporal inverted ILMFs without residual ILM peeling led to an improvement in mean vision from 113/1000 and 24/250 preoperatively to 8/20 and 71/200 one year postoperatively, without significant differences [20]. Furthermore, Lorezo et al. showed that with an MH size < 250 µm, the mean preoperative visions of 59/200 in the ILM peeling group and 62/200 in the group with inverted-ILMF with residual ILM peeling-off groups improved, with the mean final vision value being 166/200 and 204/200. In contrast, with MH sizes of 250–400 µm, mean preoperative vision values of 27/200 and 28/200 improved and were 132/200 and 123/200 six months postoperatively [29]. The final vision values after the ILM peeling and three inverted ILMF techniques were equal but depended on MH size; smaller MHs led to better postoperative vision. However, the ILM insertion resulted in the worst final vision values. In our study, mean preoperative visions values of 28/250 and 16/200 in the <200 µm and ≥200 µm groups improved to 56/200 and 35/250, 96/200 and 62/200, and 126/200 and 96/200 at one, two and three years postoperatively, respectively. Postoperative vision in both groups improved gradually and continuously from six months to three years, as long as MH closure occurred, regardless of the final lens status in phakia or pseudophakia. Better final vision was statistically related to smaller MHs, lower MH stages, better preoperative vision, ILMF not being trapped in the MH or touching the RPE one month postoperatively, MH closure found one month postoperatively, final continuous ELM and IS/OS, final good foveal contour, and a longer follow-up time. These findings are compatible with IVTS conclusions that smaller MHs with preserved outer retinal structure forecast better outcomes on SD-OCT [9].

Regarding ELM and IS/OS recovery, Michalewska et al. (MH size, >400 µm) reported that 57%/57% ELM and 24%/25% EZ defects remained one year after undergoing the classic inverted ILMF technique and temporal ILMF without residual ILM peeling, without significant differences [20]. Chou et al. (MH size, <400 µm) reported that the recovery rates for ELM and EZ reached 100%/96% and 60%/50%, respectively, after ILM peeling and temporal inverted ILMF without residual ILM peeling one year postoperatively. They also found a somewhat higher fovea-gliosis rate in the peeling group (40%) than in the flap group (20%) one month postoperatively; however, both groups had foveal gliosis around 20–30% one year postoperatively [24]. In our study, ELM and EZ recoveries in the <200 µm and ≥200 µm groups were (45.9%, 24.2%)/(19.7%, 4.8%), (48.4%, 31.3%)/(25.0%, 9.0%), and (51.6%, 29.0%)/(32.3%, 16.1%) at six months, one year, and two years postoperatively, respectively. ELM and EZ recovery rates were initially higher in the ILM peeling group than in the inverted ILMF group but plateaued later. This difference may explain distinct MH healing pathways. Müller cells after ILM peeling play a crucial role in sealing the MH [33,34] by forming gliotic tissues and non-functional photoreceptor cells to further migrate centripetally [35]. Firstly, the ELM starts recovering, the outer retinal layers continue to form, and functional recovery follows [34,35,36]. The inverted ILMF technique maintains a dry space, promoting the earlier restoration of functional retina layers and visual recovery [24]. The study predicted impending ELM and EZ recovery in the insertion group due to long-lasting glial cell proliferation [31].

In our study, in cases where the ILMF was trapped in the MH or touched the RPE one month postoperatively, the MH was relatively large (*p* <0.001), resulting in a higher final proportion of final disrupted ELM and IS/OS (*p* < 0.001) and abnormal foveal contour, including foveal gliosis and thinning (*p* = 0.003, Chi-squared test). An abnormal foveal contour including gliosis and foveal thinning was observed in approximately 15.9%/21.70% of participants, 4.3%/5.8% in the <200 µm group and 11.60%/15.90% in the ≥200 µm groups, three years postoperatively. During the early phase of MH recovery, particularly within one month postoperatively, we observed that a monolayer of ILMF touching the RPE in the MH may lead to foveal thinning and hinder functional tissue growth. However, if multilayered ILMFs are in contact with the RPE of the MH, foveal gliosis, similarly to ILM insertion, may occur. Early foveal gliosis and thinning could limit further structural changes, regardless of subsequent ELM or IS/OS recovery. However, the early establishment of a good foveal contour may lead to continuous improvement in the recovery of the outer retinal layer and vision. The potential limitations of this study include its retrospective and nonrandomized design.

## 5. Conclusions

In this retrospective cohort, the inverted ILMF technique achieved durable closure and visual gains over three years across idiopathic MH sizes. Transient flap closure and subretinal fluid may accompany MH closure with this method. Microstructural recovery (ELM, IS/OS, foveal contour) favored smaller MHs, with 20–30% non-recovery in larger holes; nonetheless, vision improved across sizes over two to three years, except when foveal gliosis or thinning was present.

## Figures and Tables

**Figure 1 diagnostics-15-02961-f001:**
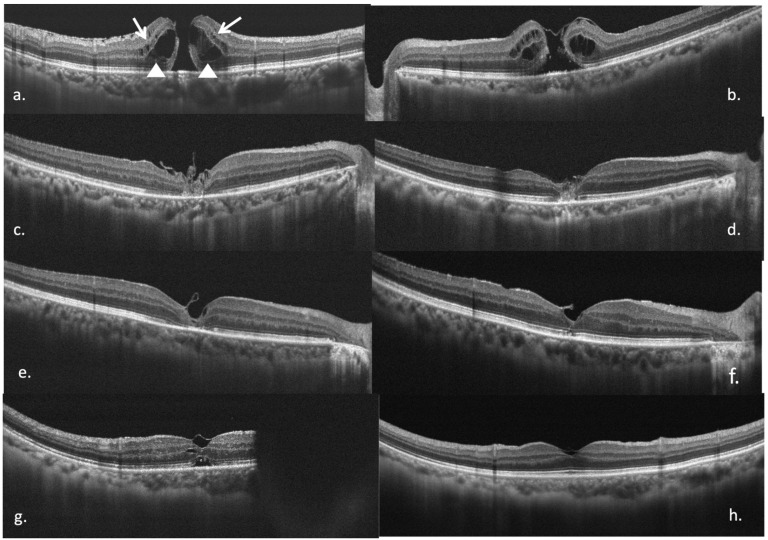
(**a**) A grade 4 full-thickness macular hole (MH) with epiretinal membrane and intraretinal cysts. Triangles represent outer retinal cysts at the outer plexiform layer and arrows represent inner retinal cysts between the inner nuclear layer and outer plexiform layer. (**b**) A case of flap closure one month postoperatively using the inverted internal limiting membrane (ILM) flap (F) technique. The ILMF covers the MH, and the two ends of the retinal tissue are kept apart. (**c**) A case of an ILMF with thick glial proliferation at the foveal area, showing homogenous hyper-reflective tissue without a continuous external limiting membrane (ELM) and inner and outer segment junction (IS/OS) one month postoperatively after undergoing the inverted ILMF technique. (**d**) One year later, the thick glial proliferation remained stationary with disrupted ELM and IS/OS. (**e**) Case of ILMF remaining in the MH and thin glial proliferation at the foveal area showing thin hyper-reflective tissue without continuous ELM and IS/OS one month after undergoing the inverted ILMF technique. (**f**) Two years later, the thin glial proliferation became thicker with continuous ELM and disrupted IS/OS. (**g**) A case of an ILMF with a thin normal foveal layer, underlying subretinal fluid, continuous ELM, and disrupted IS/OS one month postoperatively after the inverted ILMF technique. (**h**) One year later, a thin ILMF with a complete foveal contour, continuous ELM, and IS/OS was developed.

**Figure 2 diagnostics-15-02961-f002:**
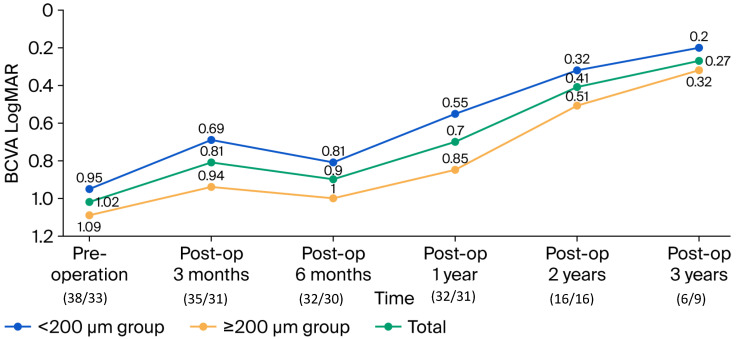
Mean logMAR best-corrected visual acuity (BCVA) changes in different groups from baseline to three years postoperatively. The mean BCVA in the <200 µm macular hole size group improved from pre-treatment to three months; however, it mildly worsened between three and six months postoperatively. From six months postoperatively, vision in the <200 µm macular hole size group vision significantly improved, while vision in ≥200 µm macular hole size group vision significantly improved from one year to two years postoperatively. (There are two counts below each time point: the first is the number of eyes with <200 µm macular hole size, the second is the number of eyes with ≥200 µm macular hole size).

**Figure 3 diagnostics-15-02961-f003:**
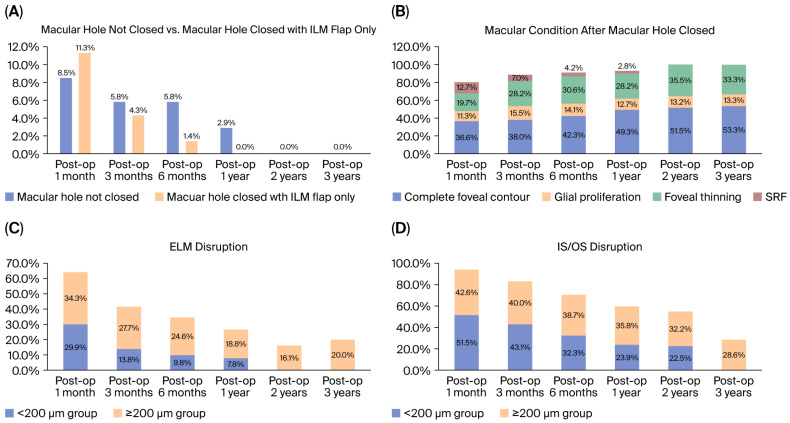
Micro-structural recovery of all eyes with an idiopathic macular hole (MH) treated with the inverted internal limiting membrane (ILM) flap (F) technique with a three-year follow-up. (**A**) ILMF closure was resolved within one year postoperatively. In the cases with MH non-closure, spontaneous closure, and another surgery was performed, enabling recovery within one year, but other cases showed no changes within two years postoperatively. (**B**) MH closure in cases with subretinal fluid was resolved within one year postoperatively. Foveal contour recovery and foveal thinning gradually improved within one year postoperatively, from 36.6%/19.7% to 49.3%/28.2% but were maintained at around 53.3%/33.3% at three years postoperatively. However, glial proliferation plateaued at around 11.3–13.3% within three years postoperatively. (**C**) Regarding external limiting membrane (ELM) recovery, the interrupted ELM was resolved in the <200-µm MH-group within two years postoperatively, while ELM was still interrupted in 16.1–20.0% of the ≥200 µm group within three years postoperatively. (**D**) For inner and outer segment junction (IS/OS) recovery, 51.5% of discontinuous IS/OS in the <200 µm MH group was totally resolved within two years postoperatively, while it remained in 28.6%% of the ≥200 µm group at three years postoperatively.

**Table 1 diagnostics-15-02961-t001:** The baseline characteristics of the study population.

Total (*n* = 71)		*n* (%)/Mean (SD)	*p*
Age (years), mean (SD)		62.38 (6.94)	0.223 ^a^
Eye (OD), *n* (%)		39 (54.93)	0.427 ^b^
Sex (Male), *n* (%)		14 (19.72)	0.256 ^b^
Spherical equivalent (D), mean (SD)		−0.16 (0.28)	0.376 ^a^
Axial length (mm), mean (SD)		23.65 (0.98)	0.177 ^a^
Lens status (phakia), *n* (%)		65 (91.55)	0.451 ^b^
Hole size (µm), mean (SD)		184.36 (94.53)	<0.001 ^a,^*
Hole stage:	stage 1 and 2, *n* (%)	24 (33.80)	0.034 ^b,^*
	stage 3 and 4, *n* (%)	47 (66.20)	
Pre-op CFT (μm), mean (SD)		393.54 (89.87)	0.361 ^a^
Pre-op vision (logMAR), mean (SD)		1.02 (0.40)	0.002 ^a,^*
Outer retinal cysts, *n* (%)		63 (88.73)	0.355 ^b^
Inner retinal cysts, *n* (%)		36 (50.70)	0.291 ^b^
Drusen, *n* (%)		23 (32.39)	0.588 ^b^
Epiretinal membrane, *n* (%)		18 (25.35)	0.582 ^b^
Post-op 1-month ILMF trapped in MH or touched RPE, *n* (%)		12 (16.90)	0.022 ^b,^*
Post-op 1-month ILMF closure, *n* (%)		8 (11.27)	0.375 ^b^
Post-op 1-month SRF, *n* (%)		9 (12.68)	0.281 ^b^
Post-op 1-month MH non-closure, *n* (%)		6 (8.45)	<0.001 ^b,^*
Final ELM (continuous), *n* (%)		53 (74.65)	<0.001 ^b,^*
Final IS/OS (continuous), *n* (%)		35 (49.30)	<0.001 ^b,^*
Final foveal contour (good), *n* (%)		43 (62.30)	<0.001 ^b,^*
glial proliferation, *n* (%)		11 (15.90)	<0.001 ^b,^*
foveal thinning, *n* (%)		15 (21.70)	<0.001 ^b,^*
Final MH non-closure, *n* (%)		2 (2.82)	0.003 ^b,^*
Complications, *n* (%)		0 (0.0)	
Final CFT (µm), mean (SD)		270.92 (47.97)	0.064 ^a^
Final vision (logMAR), mean (SD)		0.47 (0.39)	<0.001 ^c,^*
Follow-up time (month), mean (SD)		20.59 (9.87)	<0.001 ^a,^*

* *p* < 0.05, ^a^ Pearson correlation, ^b^ independent *t*-test, ^c^ paired *t*-test; *n*, number; %, percentage; SD, standard deviation; D, diopter; pre-op, preoperative; CFT, central foveal thickness; logMAR, logarithm of minimum angle of resolution; post-op, postoperative; ILMF, internal limiting membrane flap; MH, macular hole; RPE, retinal pigment epithelium; SRF, subretinal fluid; ELM, external limiting membrane; IS/OS, the junctions of the photoreceptor inner and outer segments.

**Table 2 diagnostics-15-02961-t002:** The baseline demographic data and preoperative and postoperative characteristics of the two groups.

Total: 71 Eyes	<200 µm Group	≥200 µm Group	*p*
*n* (%)	38 (53.52)	33 (46.47)	
Age (years), mean (SD)	61.37 (6.29)	63.55 (7.55)	0.189 ^b^
Eye (OD), *n* (%)	21 (29.58)	18 (25.35)	0.388 ^d^
Sex (Male), *n* (%)	8 (11.27)	6 (8.45)	0.571 ^d^
Spherical equivalent (D), mean (SD)	−0.38 (2.40)	+0.07 (2.21)	0.195 ^b^
Lens status (phakia), *n* (%)	35 (49.30)	30 (42.25)	0.841 ^d^
Hole stage (stage 1, 2), *n* (%)	32 (28.83)	8 (20.00)	0.321 ^d^
Hole size (µm), mean (SD)	134.84 (59.79)	304.19 (62.47)	<0.001 ^b,^*
Pre-op CFT (µm), mean (SD)	395.68 (101.59)	390.90 (74.62)	0.828 ^b^
Pre-op vision (logMAR), mean (SD)	0.95 (0.35)	1.09 (0.44)	0.122 ^b^
Outer retinal cysts, *n* (%)	31 (43.66)	32 (45.07)	0.060 ^d^
Inner retinal cyst, *n* (%)	16 (22.53)	37 (28.17)	0.155 ^d^
Drusen, *n* (%)	13 (18.31)	10 (14.08)	0.802 ^d^
Epiretinal membrane, *n* (%)	10 (14.08)	8 (11.27)	0.503 ^d^
Post-op 1-month ILMF trapped in MH or touched RPE, *n* (%)	3 (4.23)	9 (12.68)	0.022 ^d,^*
Post-op 1-month ILM flap closure, *n* (%)	2 (2.82)	6 (8.45)	0.026 ^d,^*
Post-op 1-month SRF, *n* (%)	8 (11.27)	1 (1.41)	0.066 ^d^
Post-op 1-month MH non-closure, *n* (%)	1 (1.41)	5 (7.04)	0.090 ^d^
Final ELM (continuous), *n* (%)	34 (49.30)	19 (27.50)	0.009 ^d,^*
Final IS/OS (continuous), *n* (%)	25 (36.20)	10 (14.50)	0.008 ^d,^*
Final foveal contour (good), *n* (%)	31 (44.90)	12 (17.40)	<0.001 ^d,^*
glial proliferation, *n* (%)	3 (4.30)	8 (11.60)	0.001 ^d,^*
foveal thinning, *n* (%)	4 (5.80)	11 (15.90)	0.001 ^d,^*
Final MH non-closure, *n* (%)	0 (0.00)	2 (2.82)	0.212 ^d^
Complications, *n* (%)	0 (0.00)	0 (0.00)	
Final CFT (µm), mean (SD)	278.58 (53.09)	262.09 (40.30)	0.150 ^b^
Follow-up time (months), mean (SD)	21.13 (10.50)	19.97 (9.20)	0.624 ^b^
Final vision (logMAR), mean (SD)	0.32 (0.30)	0.64 (0.42)	<0.001 ^b,^*
*p*	<0.001 ^c,^*	0.001 ^c,^*	

* *p* < 0.05, ^b^ independent *t*-test, ^c^ paired *t*-test, ^d^ Chi-squared test; *n*, number; %, percentage; SD, standard deviation; D, diopter; logMAR, logarithm of minimum angle of resolution; pre-op, preoperative; CFT, central foveal thickness; post-op, postoperative; ILMF, internal limiting membrane flap; MH, macular hole; RPE, retinal pigment epithelium; SRF, subretinal fluid; ELM, external limiting membrane; IS/OS, the junctions of the photoreceptor inner and outer segments.

## Data Availability

The original contributions presented in this study are included in the article. Further inquiries can be directed to the corresponding author.

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
