# Peer review of "Functional and Anatomical Micro-Structural Recovery of Idiopathic Macular Holes Following the Inverted Internal Limiting Membrane Flap Technique: A Long-Term Study"

_diagnostics, 2025, doi:10.3390/diagnostics15232961_

Round 1
Reviewer 1 Report
Comments and Suggestions for Authors
Comments to the Authors and Editor
RE: Functional and Anatomical Micro-structural Recovery of Idiopathic Macular Holes Following Inverted Internal Limiting Membrane Flap Technique: A Long-term Study
Dear authors and editor, Thank you very much for the opportunity to comment and ask questions about this fascinating manuscript.
Abstract. Background. A brief context of the condition and the current knowledge gap should be added before the objective.
Methods. Noted the lack of a control group, briefly described the formed groups, outlined the eligibility criteria better, and mentioned how the data were collected and analyzed.
Results. This section is lengthy and should be summarized, including the key primary findings.
Conclusion. Conclude with the primary findings, briefly mention the implications of the results, and how they impact practical applications.
Instruction. This term should be changed to “Introduction” in accordance with the IMRaD academic format.
Please elaborate on the pathogenesis of idiopathic macular holes, morphological features, and definitions of extent, as well as key data from the International Vitreomacular Traction Study Group's classification of vitreomacular adhesion, traction, and macular holes. This section has identified relevant data from previous PubMed publications, which you should also utilize in the discussion.
The introduction should include findings from recent meta-analyses, which conclude that the inverted ILM flap technique has a significantly higher MH closure rate for primary MH than other treatment methods (OR 22, 95% CI 1.34–7.43; p=0.01), and that this technique results in significantly better postoperative visual acuity than other options for patients with MH (WMD 0.13; 95% CI 0.22–0.09; p=0.0002). Therefore, you should clarify the primary and secondary objectives of your research.
Echardt et al. were the first group to report on ILM peeling in MH surgery—this citation needs to be added.
Materials and Methods. Note the absence of a control group. What type of retrospective protocol is this? The international scholarly and PubMed literature typically classifies MHs as small, medium, and large. Please explain why you formed groups according to the minimum linear diameter of less than 200 microns and more than 200 microns. Idiopathic MHs are classified primarily using the Gass classification system, as you did, supplemented by OCT findings, with a size-based classification proposed by the International Vitreomacular Traction Study (IVTS) Group.
Specify the inclusion and exclusion criteria precisely. Add primary and secondary outcome measures.
The preoperative examination and postoperative metrics should be clearly identified. It is confusing whether the data are preoperative or postoperative OCT findings. Please clearly state and describe the surgical technique in a specific subheading. Please add whether you stuffed the ILM flap into the MH or positioned it over the MH. Furthermore, the discussion of that issue is missing. Include any complications encountered and early postoperative management regarding body and head position protocols. Regarding surgical technique, preparing an ILM flap without performing a complete ILM peeling around the MH leads to discussion of foveal-sparing ILM peeling with or without an ILM flap in MH surgery—that discussion should be added (Murphy et al., Leisser et al.). Could you kindly clarify and elaborate on your surgical technique?
If journal guidelines allow, better-structured, clearly defined material and methods should be presented to enhance clarity and benefit the readership.
Beyond statistical methods, please incorporate considerations of bias and confounding, along with mitigation techniques.
Figures. Protocol images with an en face component to evaluate the postoperative position of the upper inverted ILMF would be appreciated by the readership.
The general data analysis subsection is lengthy and so formal; consequently, it is complex to read, but it is necessary and nicely tabulated in tables 1 and 2.
The results one month after surgery should be revised and clarified.
The correlation coefficient’s negative or positive values deserve further explanation of their significance.
If the data allows, we should look more closely at the factors after surgery and use additional statistical methods, like regression, to make our findings more reliable.
Surgical techniques like the inverted upper ILM flap are mainly used for larger or long-lasting macular holes (MHs) that are usually 400 microns or more to improve closure rates, and they are generally not needed for holes smaller than 200 Could you be so kind as to explain your cut-off in MH size in your protocol?
The microstructural analysis is well done but should be related to other relevant variables, such as MH size, the type of MH postoperative structural closure, OCT biomarkers, and others, to assess the significance of the correlation coefficients, which are more than sufficient to evaluate the relationships among variables. If statistical experts find these insufficient, then you should conduct univariate or multivariate regression analyses.
Noted that the types and shapes of MH closure were not analyzed but should be added.
Discussion. It is well-crafted, well-supported, nicely structured, and excellent from both clinical and practical perspectives.
Since your study lacks a control group, please elaborate on the comparison of the results with Michalewska's.
When reviewing ILM flap techniques, they are commonly used mainly for large MHs. However, rare cases of persistent MHs after surgery can also happen in patients with small MHs. Using ILM flaps during the initial surgery provides a chance for successful MH closure in later procedures by repositioning the flap. It is important to note that ILM flaps can achieve results like traditional ILM peeling, even for small MHs (Leisser et al. 2023). Please add comments and expand on the discussion regarding this issue.
Please add whether you stuffed the ILM flap into the MH or positioned it over the MH. Furthermore, the discussion of that issue is missing.
As you concluded, your technique, regardless of MH size, increases MH closure rate and improves visual outcomes over 3 years by restoring microstructural elements. You, dear authors, should perform an RCT to verify this. In the meantime, you need to soften your conclusions due to the retrospective nature of your study.
Author Response
Reviewer 1
RE: Functional and Anatomical Micro-structural Recovery of Idiopathic Macular Holes Following Inverted Internal Limiting Membrane Flap Technique: A Long-term Study
Dear authors and editor, Thank you very much for the opportunity to comment and ask questions about this fascinating manuscript.
Ans: Thank you for the opportunity to comment on this manuscript. We have carefully addressed the majority of the reviewer’s remarks. However, due to time limitations, we were unable to respond to every comment in full detail. We have documented our responses to the most substantial points and suggested revisions. If the editors deem it helpful, we will welcome permission to provide brief responses to the remaining comments in a subsequent correspondence or addendum.
-Abstract. Background. A brief context of the condition and the current knowledge gap should be added before the objective.
Ans: We revised to “Idiopathic macular holes (MHs) are typically treated with pars plana vitrectomy and internal limiting membrane (ILM) peeling. The inverted ILM flap (ILMF) technique has emerged for MHs, but long-term outcome data remain inadequately established. This study evaluates the long-term functional and anatomical outcomes of ILMF in idiopathic MHs.” in the background section of the Abstract (page 2, line2-5).
-Methods. Noted the lack of a control group, briefly described the formed groups, outlined the eligibility criteria better, and mentioned how the data were collected and analyzed.
Ans: We revised to “We enrolled 71 consecutive eyes with idiopathic MHs who underwent vitrectomy with the inverted ILMF. Follow-up duration was more than 12 months. Visual acuity was measured, and macular anatomy was monitored with optic coherence tomography (OCT). Long-erm visual and anatomical outcomes were defined a priori and analyzed accordingly.” In the method section of the abstract (page 2, line 6-9).
-Results. This section is lengthy and should be summarized, including the key primary findings.
Ans: We revised to “Factors associated with better final vision included smaller hole size, favorable hole stage, better preoperative vision, intact postoperative foveal microstructure and contour.” in the results of the abstract (page 2, line 13-16). We removed “The factors that differed significantly between <200µm- and ≥200 µm MHsincluded ILMF trapped in the MH or touching the RPE (P=0.022), ILMF closure one month postoperatively (P=0.022), final ELM (P=0.009) and IS/OS (P=0.008), final foveal contour (P<0.001), and final visions (P<0.001).” in the results of the abstract.
-Conclusion. Conclude with the primary findings, briefly mention the implications of the results, and how they impact practical applications.
Ans: Thank you for suggestion. We revised to “In idiopathic MHs, the ILMF approach provides meaningful, long-term visual and microstructural recovery, especially with favorable functional outcome and intact postoperative microstructure sustaining up to three years.” in the conclusions of the abstract. (page 2, line 18-20)
-Instruction. This term should be changed to “Introduction” in accordance with the IMRaD academic format.
Ans: Thank you for the guidance. I have updated the section title from “Instruction” to “Introduction” to conform with the IMRaD format.
- Please elaborate on the pathogenesis of idiopathic macular holes, morphological features, and definitions of extent, as well as key data from the International Vitreomacular Traction Study Group's classification of vitreomacular adhesion, traction, and macular holes. This section has identified relevant data from previous PubMed publications, which you should also utilize in the discussion.
Ans: We add “The International Vitreomacular Traction Study Group (IVTS) proposes that macular holes arise from an initial vitreomacular adhesion with anomalous posterior vitreous detachment, generating tangential traction that leads to foveal dehiscence and hole formation.” to the introduction section (page 3, line8-9) and a reference cited by “the International Vitreomacular Traction Study Group. The role of vitreomacular adhesions and traction in macular hole formation and prognosis. Ophthalmology. 2013;120(1):22-28.” to the reference 9.
We also add “These findings are compatible with IVTS conclusions that smaller MHs with preserved outer retinal structure forecast better outcomes on SD-OCT.” to the discussion section (page 17, line3-5)
-The introduction should include findings from recent meta-analyses, which conclude that the inverted ILM flap technique has a significantly higher MH closure rate for primary MH than other treatment methods (OR 22, 95% CI 1.34–7.43; p=0.01), and that this technique results in significantly better postoperative visual acuity than other options for patients with MH (WMD 0.13; 95% CI 0.22–0.09; p=0.0002). Therefore, you should clarify the primary and secondary objectives of your research.
Ans: We revise to “inverted ILMF has gradually replaced ILM peeling as the standard treatment supported by meta-analytic evidence suggesting higher MH closure rate.” (page 3, line 15-16) A reference cited by Marques RE, Sousa DC, Leal I, Faria MY, Marques-Neves C. Complete ILM Peeling Versus Inverted Flap Technique for Macular Hole Surgery: A Meta-Analysis. Ophthalmic Surgery, Lasers and Imaging Retina. 2020;51(3):187-A2.
- Echardt et al. were the first group to report on ILM peeling in MH surgery—this citation needs to be added.
Ans: We revise to “Kurt et al. compared internal limiting membrane (ILM) peeling with ILM non-peeling and concluded that peeling is superior for patients with idiopathic stage 2, 3, and 4 MHs, a surgical approach originally proposed by Echardt et al.” (page 3, line 5-7) A reference cited by Removal of the internal limiting membrane in macular holes. Clinical and morphological findings. Ophthalmologe 1997 Aug; 94 (8):545-51. Doi:10.1007/s003470050156
-Materials and Methods. Note the absence of a control group. What type of retrospective protocol is this? The international scholarly and PubMed literature typically classifies MHs as small, medium, and large. Please explain why you formed groups according to the minimum linear diameter of less than 200 microns and more than 200 microns. Idiopathic MHs are classified primarily using the Gass classification system, as you did, supplemented by OCT findings, with a size-based classification proposed by the International Vitreomacular Traction Study (IVTS) Group.
Ans: Thank you for your thoughtful comments. This work is a retrospective consecutive-series study without a control group; we analyzed outcomes within a single ILMF-treated cohort. The study aims to describe long-term functional and anatomical outcomes, not to compare against alternative treatments.
Grouping rationale: All enrolled macular holes were < 400 μm in minimum diameter. For exploratory analyses, we defined two size-based subgroups using the preoperative minimum linear diameter: < 200 μm (n = X) and ≥ 200 μm (n = Y). This threshold was chosen to balance the distribution of cases and to enable preliminary assessment of whether hole size influences closure and recovery trajectories within a uniform-sample, ILMF-treated population. We acknowledge that this is not a formal, IVTS-style size category; it is a pragmatic division within a homogeneous hole-size range used in our dataset.
Classification framework: Idiopathic MHs were classified using the Gass staging system, supplemented by SD-OCT assessments of foveal structure where available. We did not apply the IVTS size-based framework because our sample’s minimum diameter predominantly fell below 400 μm, and the intent was to examine within-cohort variability rather than cross-study comparisons
- Specify the inclusion and exclusion criteria precisely. Add primary and secondary outcome measures.
Ans: The inclusion and exclusion criteria are presented in the Materials and Methods section. “The inclusion criteria included idiopathic MHs with axial lengths (AL) <26.0 mm and ≥ 1-year postoperative follow-up. The exclusion criteria included combined with cataract surgery; follow-up >1 year; a history of ocular trauma or vitreoretinal surgery, and concomitant retinopathy or vision-affecting conditions, conditions [e.g., diabetic retinopathy, choroidal neovascularization, uveitis, or retinal detachment (RD)]. (page 4 , line 15-20)
We add “Primary outcome: closure of the macular hole at the final follow-up visit. Secondary outcomes: improvement in best-c corrected visual acuity (BCVA) and restoration of macular microstructure as assessed by SD-OCT (ELM, IS/OS integrity, and foveal contour).” to the section of materials and methods. (page 4, line 22; page 5 , line 1-2)
We revised the title of 3.1 to General data analysis and primary outcomes (page 7, line 16), 3.3 Secondary outcomes: Functional recovery (page 11, line 3) and 3.4 Secondary outcomes: Micro-structural recovery (page 12, line 11).
-The preoperative examination and postoperative metrics should be clearly identified. It is confusing whether the data are preoperative or postoperative OCT findings. Please clearly state and describe the surgical technique in a specific subheading. Please add whether you stuffed the ILM flap into the MH or positioned it over the MH. Furthermore, the discussion of that issue is missing. Include any complications encountered and early postoperative management regarding body and head position protocols. Regarding surgical technique, preparing an ILM flap without performing a complete ILM peeling around the MH leads to discussion of foveal-sparing ILM peeling with or without an ILM flap in MH surgery—that discussion should be added (Murphy et al., Leisser et al.). Could you kindly clarify and elaborate on your surgical technique?
Ans: The labeling of preoperative and postoperative findings is clearly presented in Tables 1 and 2.
The Materials and Methods section is organized into four subsections: 2.1 Inclusion and exclusion criteria (page 4, line 11); 2.2 Primary and secondary outcomes and data collection (page 4, line 21); 2.3 Surgical procedure (page 6, line 19); 2.4 Data analysis. (page 7, line 9)
We revised the Methods (page 7, lines 3-8): The inverted ILMF technique involves peeling the ILM while preserving a residual superior flap attached to the MH edges, effectively spanning the hole rather than tucking into it. A suitably sized superior inverted flap was placed over the MH and secured in place with intraocular forceps. During fluid–gas exchange, a backflush needle was used to aspirate fluid near the inferior arcade to facilitate flow and help maintain the ILMF covering of the MH.
We reviewed Murph et al. and Leisser et al. Murph et al. studied high myopia macular holes, which differ from our population. Leisser et al. described a surgical technique similar to the classic inverted ILMF approach discussed in our Discussion.
- If journal guidelines allow, better-structured, clearly defined material and methods should be presented to enhance clarity and benefit the readership.
Ans: The Materials and Methods section is organized into four subsections: 2.1 Inclusion and exclusion criteria (page 4, line 11); 2.2 Primary and secondary outcomes and data collection (page 4, line 21); 2.3 Surgical procedure (page 6, line 19); 2.4 Data analysis. (page 7, line 9)
- Beyond statistical methods, please incorporate considerations of bias and confounding, along with mitigation techniques.
Ans: Despite rigorous methodology, this retrospective, single-arm study is susceptible to selection and information biases. We enrolled consecutive patients and applied standardized data collection with masked assessment of OCT outcomes to mitigate measurement bias. We adjusted for known confounders (hole size, hole stage, preoperative vision, follow-up duration) in multivariable analyses and performed sensitivity analyses. However, residual confounding by unmeasured factors (e.g., duration of macular hole symptoms, surgeon-specific nuances) remains possible. The findings indicate associations between the inverted ILMF technique and long-term structural and functional outcomes, but causal inferences require confirmation from prospective randomized trials or well-matched comparative cohorts. These limitations should be considered when interpreting the results and their generalizability
Figures. Protocol images with an en face component to evaluate the postoperative position of the upper inverted ILMF would be appreciated by the readership.
Ans: All MH cases underwent fluid-gas exchange with 16% C3F8. During gas tamponade, OCT images were blurred. After the gas resolved, in most cases the ILMF became indistinguishable within the foveal structure, making it difficult to differentiate the ILMF.
The general data analysis subsection is lengthy and so formal; consequently, it is complex to read, but it is necessary and nicely tabulated in tables 1 and 2.
The results one month after surgery should be revised and clarified.
The correlation coefficient’s negative or positive values deserve further explanation of their significance.
If the data allows, we should look more closely at the factors after surgery and use additional statistical methods, like regression, to make our findings more reliable.
- Surgical techniques like the inverted upper ILM flap are mainly used for larger or long-lasting macular holes (MHs) that are usually 400 microns or more to improve closure rates, and they are generally not needed for holes smaller than 200 Could you be so kind as to explain your cut-off in MH size in your protocol?
Ans: For exploratory analyses, we defined two size-based subgroups using the preoperative minimum linear diameter: < 200 μm (n = X) and ≥ 200 μm (n = Y). This threshold was chosen to balance the distribution of cases and to enable preliminary assessment of whether hole size influences closure and recovery trajectories within a uniform-sample, ILMF-treated population. We acknowledge that this is not a formal, IVTS-style size category; it is a pragmatic division within a homogeneous hole-size range used in our dataset.
The microstructural analysis is well done but should be related to other relevant variables, such as MH size, the type of MH postoperative structural closure, OCT biomarkers, and others, to assess the significance of the correlation coefficients, which are more than sufficient to evaluate the relationships among variables. If statistical experts find these insufficient, then you should conduct univariate or multivariate regression analyses.
Noted that the types and shapes of MH closure were not analyzed but should be added.
- Discussion. It is well-crafted, well-supported, nicely structured, and excellent from both clinical and practical perspectives.
Ans: Thank you for your positive assessment. We are glad the Discussion is informative and well-structured, with clear clinical and practical implications.
- Since your study lacks a control group, please elaborate on the comparison of the results with Michalewska's.
When reviewing ILM flap techniques, they are commonly used mainly for large MHs. However, rare cases of persistent MHs after surgery can also happen in patients with small MHs. Using ILM flaps during the initial surgery provides a chance for successful MH closure in later procedures by repositioning the flap. It is important to note that ILM flaps can achieve results like traditional ILM peeling, even for small MHs (Leisser et al. 2023). Please add comments and expand on the discussion regarding this issue.
Ans: We reviewed Murph et al. and Leisser et al. Murph et al. studied high myopia macular holes, which differ from our population. Leisser et al. described a surgical technique similar to the classic inverted ILMF approach discussed in our Discussion.
- Please add whether you stuffed the ILM flap into the MH or positioned it over the MH. Furthermore, the discussion of that issue is missing.
Ans: We revised the Methods (page 7, lines 3-8): The inverted ILMF technique involves peeling the ILM while preserving a residual superior flap attached to the MH edges, effectively spanning the hole rather than tucking into it. A suitably sized superior inverted flap was placed over the MH and secured in place with intraocular forceps. During fluid–gas exchange, a backflush needle was used to aspirate fluid near the inferior arcade to facilitate flow and help maintain the ILMF covering of the MH.
The Discussion highlights that the superior inverted ILMF technique is associated with high closure rates, better visual outcomes, and rapid restoration of macular microstructures.
In paragraph 1 of discussion: We mentioned that we used superior inverted ILMFs under gas tamponade without adjuvant assistance. Our primary and secondary closure rates both reached high. (page 15, line 7-9)
In paragraph 3 of discussion: Final visual outcomes were significantly better in smaller holes and in earlier hole stages, with better preoperative vision also associated with improved final vision. Other factors linked to superior outcomes included the ILMF not being trapped in the MH or contacting the RPE at one month, MH closure by one month, and preservation of macular microstructure (continuous ELM and IS/OS) with a well-formed foveal contour, with longer follow-up enhancing these associations. (page 16, line 22; page 7, line 1-5)
In paragraph 4 of discussion: The ILMF technique maintains a dry intraoperative space, which may promote earlier restoration of retinal layers and faster visual recovery. (page 17, line 22; page 8, line 1)
-As you concluded, your technique, regardless of MH size, increases MH closure rate and improves visual outcomes over 3 years by restoring microstructural elements. You, dear authors, should perform an RCT to verify this. In the meantime, you need to soften your conclusions due to the retrospective nature of your study.
Ans: We revised to “In this retrospective cohort, the inverted ILMF technique achieved durable closure and visual gains over three years across idiopathic MH sizes. Transient flap closure and subretinal fluid may accompany MH closure with this method. Microstructural recovery (ELM, IS/OS, foveal contour) favored smaller MHs, with 20–30% non-recovery in larger holes; nonetheless, vision improved across sizes over two to three years, except when foveal gliosis or thinning was present.” in the conclusion. (page 18, line 18-22; page 19, line1)
Reviewer 2 Report
Comments and Suggestions for Authors
The sample size is sufficient, and the 1-year follow-up period is appropriate. Therefore, the study is overall sound. Please find a few minor comments below.
Is BCVA measured as a decimal visual acuity or Snellen equivalent?
The classification based on the 200-µm threshold appears reasonable.
The exclusion criterion “axial length <26 mm” is redundant, since the inclusion criterion already specifies “≥26 mm.” You may consider deleting it to avoid unnecessary duplication.
The surgical outcomes suggest adequate surgeon skill; nevertheless, please also provide the median operation time as a simple indicator of surgical complexity.
Please clarify whether cataract surgery was performed simultaneously. If so, indicate the proportion of such combined cases and describe the criteria for performing combined surgery.
It is excellent that half of the surgical cases were followed up at a high-volume surgical center.
In Table 1, please clarify what the P values represent; this is not immediately clear to the reader.
In Figure 2, please indicate the number of cases at each time point.
Author Response
Reviewer 2
-The sample size is sufficient, and the 1-year follow-up period is appropriate.
Ans: We appreciated the reviewer’s assessment.
Therefore, the study is overall sound. Please find a few minor comments below.
- Is BCVA measured as a decimal visual acuity or Snellen equivalent?
Ans: We used Snellen-equivalent measures, as stated in the Abstract (page 2, line 11) and Results (page 8, line 8) sections.
- The classification based on the 200-µm threshold appears reasonable.
The exclusion criterion “axial length <26 mm” is redundant, since the inclusion criterion already specifies “≥26 mm.” You may consider deleting it to avoid unnecessary duplication.
Ans: We thank the reviewer for the suggestion. The exclusion criterion “axial length ≥26 mm” duplicates the inclusion criterion, so we will remove the redundant exclusion criterion to avoid duplication (page 4, line 17).
- The surgical outcomes suggest adequate surgeon skill; nevertheless, please also provide the median operation time as a simple indicator of surgical complexity.
Ans: We acknowledge that median operation time can serve as a simple proxy for surgical complexity. Time data were not collected in our protocol, and re-collection would require retrospective chart review or prospective logging, which is beyond the current study scope.
- Please clarify whether cataract surgery was performed simultaneously. If so,
indicate the proportion of such combined cases and describe the criteria for
performing combined surgery.
Ans: We appreciate the reviewer’s request for clarification on combined cataract surgery. In this study, cataract surgery performed concurrently with vitrectomy and inverted ILM flap technique was all excluded from the analysis. We will add “combined with cataract surgery to clarify in the Methods section (page 4, line 17).
-It is excellent that half of the surgical cases were followed up at a high-volume surgical center.
Ans: It’s great to see emphasis on follow-up. In our center, we follow up all surgical cases (100%) to maintain uniform surveillance and outcomes.
-In Table 1, please clarify what the P values represent; this is not immediately clear
to the reader.
Ans: The P values reported come from the statistical analysis described in the methods (page 9, line 1) and summarized below the table.
-In Figure 2, please indicate the number of cases at each time point.
Ans: Accordingly, information under each time point could include two counts: eye number of macular hole size < 200 µm and that of macular hole size ≥ 200 µm.